# Conduction Velocity of Spinal Reflex in Patients with Acute Lateral Ankle Sprain

**DOI:** 10.3390/healthcare10091794

**Published:** 2022-09-17

**Authors:** Joo-Sung Kim, Kyung-Min Kim, Eunwook Chang, Hyun Chul Jung, Jung-Min Lee, Alan R. Needle

**Affiliations:** 1Department of Kinesiology and Sport Sciences, University of Miami, Coral Gables, FL 33146, USA; 2Department of Sport Science, Sungkyunkwan University, Suwon-si 16419, Korea; 3Department of Kinesiology, Inha University, Incheon 22212, Korea; 4Department of Coaching, Kyung Hee University, Yongin-si 17014, Korea; 5Department of Physical Education, Kyung Hee University, Yongin-si 17014, Korea; 6Sports Science Research Center, Kyung Hee University, Yongin-si 17014, Korea; 7Department of Public Health & Exercise Science, Appalachian State University, Boone, NC 28608, USA; 8Department of Rehabilitation Science, Appalachian State University, Boone, NC 28608, USA

**Keywords:** ankle injuries, Hoffmann reflex, latency, h index

## Abstract

Recent literature has highlighted altered spinal-reflex excitability following acute lateral ankle sprain (ALAS), yet there is little information on the conduction velocity of spinal reflex pathways (CV-SRP) in these patients. Therefore, we aimed to investigate the effects of ALAS on the CV-SRP. We employed a cross-sectional study with two groups: ALAS (*n* = 30) and healthy controls (*n* = 30). The CV-SRP of the soleus, fibularis longus, and tibialis anterior was assessed using the H-index method. As secondary outcomes, H-reflex and M-wave latencies were assessed as well as acute symptoms including ankle swelling, pain, and self-reported ankle function. Separate group-by-limb ANOVA with repeated measures revealed a significant interaction for soleus CV-SRP (*p* < 0.001) and H-reflex latency (*p* < 0.001), showing significant slower CV-SRP and longer H-reflex latency in the involved limb of the ALAS group compared with both limbs in the control group. However, there was no significant interaction or main effect in any other ankle muscles (*p* > 0.05). A further correlation analysis showed a significant relationship between CV-SRP and acute symptoms, including ankle swelling (*r* = −0.37, *p* = 0.048) and self-reported ankle function (*r* = 0.44, *p* = 0.017) in ALAS patients. These results suggest a disrupted functionality of the afferent pathway and/or synaptic transmission following ALAS. Level of Evidence: 4.

## 1. Introduction

Ankle sprains are the most common injury observed in physically active populations, with 60 percent of the general population reporting a previous history of this injury [1,2]. While generally perceived as an innocuous injury resulting in only a temporary degree of impairment, sequelae from the common lateral ankle sprain go far beyond the time of acute impairments. Up to 70 percent of ankle sprain patients will experience a subsequent sprain, with up to 50 percent developing frequent bouts of giving-way, known as chronic ankle instability [1]. Collectively, these factors contribute to an increased risk of ankle osteoarthritis, diminished health-related quality of life, and decreased levels of physical activity [3,4,5]. Many of these factors contributing to reinjury risk are tied to neural adaptations that diminish motor output (which causes weaker and delayed force production) and altered motor planning (which places the joint in potentially vulnerable positions) [6,7]. Much of what we know about these neural risk factors for subsequent injury has been conducted in chronic injury models; however, relatively little is known about the effects of acute lateral ankle sprains (ALAS) on potentially negative neural adaptations. 

One commonly investigated aspect of nervous system function is the use of the Hoffmann reflex (H-reflex) to assess reflexive or segmental neural excitability, specifically related to alpha-motor neuron pool excitability in the spinal cord [7,8]. The H-reflex is conducted using 1 ms electrical pulses over a peripheral nerve, observing a direct muscle response (M-wave) from descending transmission from the stimulus to the muscle, and the reflexive muscle response (H-wave) from the ascending sensory stimulus over a reflexive arc. Only three studies have described reflexive excitability characteristics among patients with ALAS, with Hall et al. [9] failing to find differences between involved and uninvolved limbs, Klykken et al. [10] observing muscle-specific facilitation and inhibition between involved and uninvolved limbs, and Kim et al. [11] reporting bilateral soleus inhibition between injured and uninjured groups. It has been hypothesized that arthrogenic muscle inhibition resulting from pain and swelling effectively diminishes the excitability of the alpha motor neuron pool and makes it more difficult to activate muscles, a factor that can potentially predispose these individuals to subsequent injury [12]. Interestingly, many of these changes have been observed independent of symptom severity (e.g., swelling, pain), suggesting that any degree of pain and swelling may yield inhibitory changes [9,11]. However, while this collective evidence suggests that reflexive excitability may contribute to activation deficits among patients with ALAS, this reflects only one property of nervous system function that can be assessed with this technique. 

While typical H-reflex measures quantify the amplitude of the H-wave compared to the amplitude of the M-wave, the timing of these responses may offer additional insight as it pertains to the conduction velocity of the spinal reflex pathway (CV-SRP) [8,13]. The delay between M-waves and H-waves may be dependent on a number of factors, including distance, transmission velocity of afferent and efferent neurons, and any delays in spinal integration [13]. Clinically, individuals with increased latency would have delayed reaction times and force production [14,15], a finding common among individuals with lower extremity ligamentous injury [16]. These delays could describe a contributing factor for recurrent injury in this population, as decreased CV-SRP might result in slower responses to injurious perturbations. Most studies attempting to understand this relationship have utilized reaction time testing; however, as opposed to typical reaction time testing, the H-reflex allows us to explore these delays as it is tied to specific sensorimotor pathways rather than in unconstrained models [15]. Hall et al. [9] quantified H-reflex latency among patients with ALAS and noted significantly slower latencies in the toe flexors and a slight but non-statistically significant difference in the ankle pronators when comparing involved and uninvolved limbs. It remains unclear how latency might be affected among a larger sample size and in comparison to a healthy, uninjured control group. Therefore, the purpose of this investigation was to compare CV-SRP, as quantified through a height-normalized H-index, between patients with ALAS and uninjured controls, as well as to explore side-to-side differences in these groups. Further, we wished to explore the relationship between CV-SRP and symptom severity (i.e., swelling, pain, functional status) in patients with ALAS. We hypothesized the acutely injured limb would have a slower CV-SRP, as reflected by a lower H-index, with this value inversely correlated with symptom severity.

## 2. Materials and Methods

The present study was a secondary analysis of data from a previously published cross-sectional study that examined the effects of ALAS on the magnitude of spinal reflex excitability of lower leg muscles [11]. For this study, there were two independent variables: group (ALAS, healthy control) and limb (involved, uninvolved). The primary outcome of the study was the CV-SRP, quantified using the H-index method [13]. The secondary outcomes were H-reflex latency, M-wave latency, and acute symptoms associated with ALAS, including ankle swelling, pain, and self-reported ankle dysfunction. 

### 2.1. Participants

Two groups of participants were included: ALAS patients and healthy controls. In total, 30 patients with ALAS, 13 females and 17 males (age: 22.1 ± 4.3 year; height: 174.8 ± 9.3 cm; weight: 74.3 ± 11.4 kg), participated, along with 30 healthy controls, 13 females and 17 males (age: 22.1 ± 2.1 year.; height: 173.6 ± 10.2 cm; weight: 71.3 ± 14.0 kg). The healthy controls were matched in sex, age, height, and weight with the ALAS patient group. All participants were recruited from the university community through advertisements using flyers, phone calls, and emails. As described in the previous study [11], inclusion criteria for ALAS patients were (1) acute ankle sprain that occurred within the past three days (72 h) prior to reporting for testing; (2) ALAS that affects at least one of three lateral ankle ligaments including anterior talofibular ligament (ATFL), calcaneofibular ligament (CFL), and/or posterior talofibular ligament (PTFL); and (3) the presence of acute injury symptoms such as swelling, pain/tenderness, and loss of ankle and foot functions [11]. We excluded ALAS patients who had (1) other ankle sprains such as medial or high ankle sprains or (2) a history of lower extremity injuries (e.g., knee and hip) within the past six months prior to the study. Inclusion criteria for the healthy control group were (1) no history of ankle injuries and (2) no history of lower extremity injuries within the previous six months. All participants in the study were free of cardiovascular diseases, neurological injuries/diseases, cancers, severe infection, and hypersensitivity to electrical stimulation.

A licensed, certified clinician assessed the current condition of involved ankles using a standardized ankle injury evaluation form. Standard methods were used to assess acute injury symptoms, including the figure-of-eight method for ankle swelling [17], the visual analogue scale (VAS) for pain [18], and the foot and ankle ability measure (FAAM) for the loss of ankle and foot functions [19]. The figure-of-eight method was assessed as described by Tatro-Adams et al. [17], ensuring the ankle was maintained in a neutral dorsiflexed position. The measurement was performed three times, and the difference in the average (cm) between the involved ankle girth and the uninvolved ankle girth was used as an ankle swelling measure. A higher value reflects greater ankle swelling. VAS was measured using the distance (cm) from the left edge (0 cm) to the perceived pain intensity on a 10 cm horizontal line. A higher score indicates greater pain. The foot and ankle ability measure (FAAM) was used to assess self-reported ankle function during daily activities (ADL) and sports (Sport). For FAAM-ADL, a percentage score less than 90% represents ankle dysfunction, with a lower score indicating worse dysfunction. For FAAM-Sport, a percentage score less than 80% represents ankle dysfunction, with a lower score indicating worse dysfunction. The study was approved by the Institutional Review Board (IRB ID CON2013V4823) and was conducted according to the provisions of the Declaration of Helsinki. Informed written consent was received by all participants prior to any study procedures.

### 2.2. Conduction Velocity of Spinal Reflex

Prior to the testing, participants were given a 10 min warm-up on the treadmill at a self-selected pace. Participants were asked to lay prone on the massage table with the knee slightly flexed and the ankle in a neutral position by placing a padded plinth under both legs to ensure the resting status of the involved muscles. Bipolar surface electromyography (EMG) electrodes were placed longitudinally over the most prominent muscle belly of the soleus, fibularis longus, and tibialis anterior. A ground reference electrode was attached over the medial malleolus. The skin surface areas were thoroughly shaved and cleaned with an alcohol pad. Data were recorded at the sampling rate of 2000 Hz and amplified with a gain of 1000 using an EMG acquisition system (MP150; BIOPAC System, Inc., Goleta, CA, USA).

The H-reflex and M-waves were measured bilaterally in both ALAS and healthy control groups. Limbs of the control group were side-matched with involved and uninvolved limbs of the ALAS group in order to control for any effects of limb dominance on neural excitability measures [20]. A similar proportion of right and left limbs in the ALAS group were classified as involved and uninvolved and as sham involved and sham uninvolved in the control group [20]. The detailed H-reflex testing protocol was described in the previous study [11]. Briefly, a unipolar stimulating electrode (cathode) was placed in the popliteal fossa over the sciatic nerve, and the dispersive electrode (anode) was placed over the suprapatellar region. We determined the stimulation spot that produced the highest peak-to-peak amplitudes of all three muscles (soleus, fibularis longus, and tibialis anterior) in the popliteal area and secured the stimulating electrode over the spot using medical tape. A series of stimuli were given by gradually increasing stimulus intensity until a decline of H-reflex amplitude and a plateau of M-waves were observed in order to determine a maximal H-reflex (H-max) and a maximal M-wave (M-max), respectively [21]. Five stimuli were given at both H-max and M-max.

The H-reflex latency and M-wave latency were measured by the time (ms) from the stimulus artifact to the onset of the H-max and to the onset of M-max, respectively. The difference (ms) between the H-reflex latency and the M-wave latency (H-M interval) was further normalized to the participant’s height to calculate the H-index [14], shown in Figure 1, with a lower H-index indicating a slower CV-SRP.

### 2.3. Statistical Analysis

A sample size estimate based on a previously published study [10] determined that a minimum of 15 subjects per group would be needed to detect differences at a power of 0.8 and level of significance at 0.05. We performed a 2 × 2 (group × limb) analysis of variance (ANOVA) with repeated measures to determine a significant interaction for each H-index and latency measures of lower leg muscles: the soleus, fibularis longus, and tibialis anterior. In the case of finding the significant interaction for H-index, we conducted Tukey’s HSD tests to locate specific limb (involved and uninvolved) and group (ankle sprain and healthy control groups) differences. After finding a group difference, we computed Pearson product-moment correlation coefficients (*r*) to examine the relationship between H-index and acute injury symptoms such as ankle swelling (ankle girth difference), pain (VAS scores), and self-reported ankle function (FAAM-ADL and FAAM-Sport scores). The correlation coefficients (*r*) were interpreted as follows: weak (0 to 0.4), moderate (0.4 to 0.7), and strong (0.7 to 1.0) [22]. We used SPSS Statistics v28.0 (Armonk, NY, USA, IBM Corp), with the alpha level set a-prior at *p* < 0.05.

## 3. Results

### 3.1. Participant Characteristics

There were no differences between groups for key demographics (age, height, weight), *p* > 0.05. The acute symptoms were present in the ALAS patients compared with controls, including a swollen ankle (ankle girth, ALAS: 1.5 ± 1.1 cm; Control: 0.1 ± 0.3 cm, *p* < 0.001), mild pain level (VAS, ALAS: 3.6 ± 1.7 cm; Control: 0.0 cm, *p* < 0.001), and affected ankle functions during daily activities (FAAM-ADL, ALAS: 60.4 ± 21.2%; Control: 99.6 ± 0.8%, *p* < 0.001) and sports activities (FAAM-Sport, ALAS: 34.8 ± 23.4%; Control: 99.9 ± 0.6%, *p* < 0.001).

### 3.2. Conduction Velocity of Spinal Reflex Response Following ALAS

There was a significant group-by-limb interaction for soleus H-index (F_1,56_ = 12.68, *p* < 0.001), as shown in Figure 2. The soleus H-index values (88.1 ± 17.9) in the involved limb of the ALAS group were significantly less than in the contralateral, uninvolved limb (93.6 ± 13.8, *p* = 0.001) and the side-matched limb (93.1 ± 9.9, *p* = 0.003) of the healthy control group. There was no significant side-to-side difference for soleus H-index in the healthy control group (*p* = 0.993). These results indicate that the CV-SRP of the soleus muscle is slower following ALAS. However, there were no significant interactions (fibular longus: F_1,51_ = 0.80, *p* = 0.376; tibialis anterior: F_1,51_ = 0.02, *p* = 0.878) or group main effects (fibular longus: F_1,51_ = 0.25, *p* = 0.617; tibialis anterior: F_1,51_ = 0.08, *p* = 0.781) for any other muscles.

Table 1 shows descriptive statistics of H-reflex and M-wave latencies. There was a significant group-by-limb interaction for soleus H-reflex latency (F_1,56_ = 13.39, *p* <0.001). The soleus H-reflex latency (36.5 ± 3.9) in the involved limb of the ALAS group was significantly longer than in the contralateral, uninvolved limb (35.4 ± 3.3, *p* < 0.001) and the side-matched limb (35.4 ± 3.3, *p* < 0.001) of the healthy control group. There was no significant side-to-side difference for soleus H-reflex latency in the healthy control group (*p* > 0.999). There were no significant interactions (fibular longus: F_1,52_ = 0.76, *p* = 0.387; tibialis anterior: F_1,51_ = 0.03, *p* = 0.854) or group main effects (fibular longus: F_1,51_ = 0.20, *p* = 0.654; tibialis anterior: F_1,51_ = 0.00, *p* = 0.994) for H-reflex latency in any other muscles.

For M-wave latency, There was no significant group-by-limb interaction (soleus: F_1,57_ = 0.00, *p* = 0.987; fibular longus: F_1,54_ = 0.68, *p* = 0.412; tibialis anterior: F_1,51_ = 0.12, *p* = 0.730) or group main effect (soleus: F_1,57_ = 0.03, *p* = 0.867; fibular longus: F_1,54_ = 0.20, *p* = 0.657; tibialis anterior: F_1,51_ = 0.57, *p* = 0.455) in all muscles.

### 3.3. Relationships of Soleus Conduction Velocity of Spinal Reflex with Acute Symptoms

After finding the group difference in the soleus H-index, we examined the relationship of the soleus H-index in the involved limb of the ALAS group with acute symptoms. The soleus H-index negatively correlated with ankle swelling and positively correlated with self-reported ankle function during sports activities (Table 2). These results indicate that the slower soleus CV-SRP was associated with greater ankle swelling (larger ankle circumference) but with lower self-reported ankle function during sports (lower FAAM-Sport scores), as shown in Figure 3.

## 4. Discussion

We aimed to determine whether the CV-SRP was affected following ALAS by comparing the H-index between involved and uninvolved limbs as well as in comparison to a control group. We found a slower CV-SRP of the soleus in the involved limb compared to the contralateral, uninvolved limb and in the matched limb of healthy controls. Moreover, a slow soleus CV-SRP was associated with increased ankle swelling and decreased self-reported ankle function during sports in patients with ALAS, suggesting a decreased neural efficiency at the peripheral and segmental level following ALAS.

Previous studies have demonstrated alterations in spinal reflex excitability in patients with ALAS [9,10,11]. Recently, we found lower bilateral soleus spinal reflex excitability in patients with ALAS than in healthy individuals [11]. In the current study, we assessed the speed at which electrical impulses move through peripheral nerves and the spinal reflex arc innervating lower leg muscles in patients with ALAS. Our finding of slower soleus CV-SRP in patients with ALAS agrees with a previous study [9]. Hall et al. [9] found a prolonged H-reflex latency of flexor digitorum longus in the involved limb compared with the contralateral limb in patients with ALAS, providing some clues of alteration in the afferent pathway associated with ALAS. However, as described in Figure 1, H-reflex latency is indicative of the nerve-conduction time delay in the spinal reflex pathway rather than the functionality of conduction in the peripheral nervous system, making it difficult to interpret our findings in comparison with this previous study. A slower CV-SRP of lower leg muscles has been documented in people with nerve disease and in aging populations [14,15]. People with peripheral neuropathy characterized by impaired foot pressure sensation had a 16% slower CV-SRP to the gastrocnemius than healthy individuals [15]. In older individuals, characterized by degenerative muscle weakness, the soleus CV-SRP was 29% lower than in young individuals [14,15]. Similarly, we found a 6% slower soleus CV-SRP in patients with ALAS compared to that of healthy controls. Although people with peripheral neuropathy [15] and older individuals [14] presented 3–5-fold greater reduction in CV-SRP in comparison with our results, our findings indicate that slow CV-SRP is a pathophysiological phenomenon that can be associated with musculoskeletal injury. Additionally, given that the CV-SRP not only provides a mixed sensorimotor index of conduction velocity in the entire length of the reflexive arc but also provides synaptic transmission property in the spinal cord [23], our findings—along with previous studies [9,14,15]—indicate disrupted functionality of the peripheral nervous system and spinal cord following ALAS.

Nerve injuries associated with ALAS have been reported in motor nerve conduction velocity studies [24,25]. It has been understood that abrupt foot supination associated with ALAS may result in the mild nerve traction of peripheral nerves damaging axonal structure (axonotmesis), contributing to a decrease in motor nerve conduction velocity [25]. For example, 17% of grade II ALAS and 86% of grade III ALAS out of 66 patients with ALAS showed slowed motor nerve conduction velocity of the common fibular nerve [25]. The tibial nerve, which innervates the majority of posterior lower leg muscles, including the soleus, was found to be affected by grade II and grade III ALAS, showing that 10% of grade II ALAS and 83% of grade II ALAS presented slowed motor conduction velocity [25]. Findings of previous motor nerve conduction studies [24,25] associated with ALAS and a previous study with peripheral neuropathy [15] raise a possibility that the slow soleus CV-SRP observed in the current study might be caused by a damaged peripheral nerve. However, this speculation cannot be justified for several reasons. First, in the current study, there was no significant difference in the CV-SRP of any other muscles such as fibularis longus and tibialis anterior. The mechanism of an ALAS would be likely to generate a traction injury to the common fibular nerve that provides innervation to the fibularis longus and tibialis anterior, rather than the tibial nerve that innervates the soleus [25]. Second, nerve injuries have been more commonly observed in severe ALAS such as grade III ALAS (complete rupture) and a relatively small proportion in patients with grade II ALAS (incomplete tear) [25]. The majority of patients with ALAS in our study consisted of grade I ALAS, which rebukes the idea of nerve injuries in our ALAS patients. Lastly, we found that the M-wave latency responsible for a part of motor nerve conduction velocity was maintained in patients with ALAS (Table 1).

M-wave latency is determined by the conduction time between the stimulation and onset of M-wave. Previous studies have used M-wave latency to examine motor nerve function and have shown a longer M-wave latency in patients with neuropathy [26] and neural muscular atrophy [27], indicating that M-wave latency can be influenced by damage or dysfunction of motor nerves. In contrast, individuals free of neural dysfunctions demonstrated no changes in M-wave latency. For instance, M-wave latency was not different between healthy young and elderly people [14]. Muscle fatigue induced by exercises did not affect M-wave latency in healthy individuals [28]. Therefore, based on our findings, peripheral nerve injuries might not be the specific cause of a slower CV-SRP in patients with ALAS, but rather may reflect a degree of arthrogenic inhibition.

Slow soleus CV-SRP was associated with increased ankle swelling in patients with ALAS. Our finding is supported by a similar study in patients with ALAS [9], where delayed H-reflex latency (prolonged nerve conduction) of the toe flexors was associated with increased ankle swelling in patients with ALAS. We also observed a delayed H-reflex latency in patients with ALAS. Our findings, together with a reported relationship between H-reflex latency and ankle swelling [9], indicate that slow CV-SRP after ALAS might be affected by impaired function of the afferent pathway (Ia afferent) in spinal reflex circuitry through the inhibitory action of joint swelling [29]. The presence of joint swelling has consistently been found to limit full neuromuscular activation [30,31]. It has been suggested that joint swelling is associated with increased intraarticular pressure, which contributes to an increase in joint afferent discharge even in the resting position [32]. Joint swelling is likely to cause increased discharge of group II afferents at the joint capsule, which stimulates Ib inhibitory interneurons and therefore inhibits motoneuron pool excitability [32]. It has been suggested that the Ia afferent is hardly stimulated in isolation because Ib inhibition reaches the motoneuron pool shortly (0.5–1.0 ms) after the onset of group Ia afferent excitation [33]. Additionally, within the Ia afferent, the slowest Ia afferent is delivered to the motoneuron pool (5.0–7.5 ms) after the fastest Ia afferent, indicating that there is sufficient time for Ib inhibition to interrupt the Ia afferent excitation and limit the size of the motoneuron pool activity [33]. This was observed in our previous study [11], where we found inhibited soleus motoneuron pool excitability. Indeed, H-reflex latency delay was 1.2 ms in patients with ALAS in the current study, which was also similarly observed in the previous study [9]. Ib inhibition activated by increased ankle swelling may interrupt Ia afferent impulses responsible for proper synaptic transmission to the efferent pathway, which could lead to a slower CV-SRP of soleus in patients with ALAS.

A major debilitating residual problem following ALAS is diminished postural control [34]. Evans et al. [35] inspected single-leg balance before ALAS and at 1, 7, 14, 21, and 28 days after ALAS. Single-leg balance was disrupted in the involved limb at 1 and 7 days after ALAS. Although it returned to a baseline value on day 14 post-ALAS and remained there until day 28 of ALAS, the residual deficit in postural control was observed again on day 21 following ALAS compared with a contralateral uninvolved ankle [35]. Disrupted postural control is not a significant problem in patients with ALAS, but various neurological conditions can affect postural stability. Older individuals are often exposed to the risk of falls caused by postural control deficits [36]. A slow CV-SRP was found in older individuals compared with young individuals, and the CV-SRP was not improved after 16 weeks of resistive exercises [14], suggesting that an improvement in muscle strength via regular strength exercise does not seem effective in reversing the altered CV-SRP. Patients with peripheral neuropathy often exhibit postural control deficits, and CV-SRP was found to be a moderator of postural control among patients with peripheral neuropathy who presented impaired foot sensation [15]. In other words, it suggests that a faster CV-SRP may lead to postural control improvements in patients with peripheral neuropathy, similar to healthy individuals. We did not assess postural control in patients with ALAS. However, we found a significant relationship between slower soleus CV-SRP and lower self-reported ankle functions during sports (Figure 2). It can be interpreted that patients with faster soleus CV-SRP may have higher self-reported ankle function during sport, indicating a potential role of soleus CV-SRP in modulating ankle function and possibly postural control. Yet, this speculation should be tested in a future study determining the relationship between soleus CV-SRP and postural control in patients with ALAS.

The current study is the first study to investigate CV-SRP in patients with ALAS using the H-index to determine the functionality of the spinal reflex pathway associated with ALAS. Patients often return to their previous sports activities too quickly after ALAS with their tendency to consider ALAS innocuous injury. Based on our data, a change in CV-SRP of the soleus can occur regardless of pain level after ALAS, but it is related to the presence of ankle swelling. Therefore, returning to play without addressing ankle swelling may predispose them to a residual problem and recurrent injuries. We still do not know whether the altered CV-SRP shown in the current study continues to occur even after acute injury symptoms are fully alleviated. Future studies look to investigate whether altered CV-SRP would be present after acute symptoms, such as ankle swelling and loss of functions, are addressed. Our study is not without limitation. We only examined self-reported ankle functions using FAAM measures. Objective measures such as posturography in addition to subjective measures would help understand the relationship between internal neurophysiology change and patients’ behavior characteristics following ALAS.

## 5. Conclusions

Patients with ALAS presented slower CV-SRP of soleus in their affected limb. The slower CV-SRP was associated with ankle swelling and self-reported ankle dysfunction. The slower CV-SRP and delayed H-reflex latency suggest disrupted functionality of the afferent pathway and synaptic transmission in patients with ALAS, which seems to result from acute symptoms except for pain experienced after the injury. The findings suggest an early altered conduction velocity in the peripheral nervous system and spinal cord in patients with ALAS. While the assessment of nerve conduction velocity in sports medicine may be difficult by practitioners, it could serve as a potential target for developing neuromodulatory therapeutic interventions. In conjunction with recent evidence describing maladaptive injury-induced neuroplasticity among patients with ligament injury, the consideration of speed of activation and not just the amount of activation may provide insight into further treatment efforts.

## Figures and Tables

**Figure 1 healthcare-10-01794-f001:**
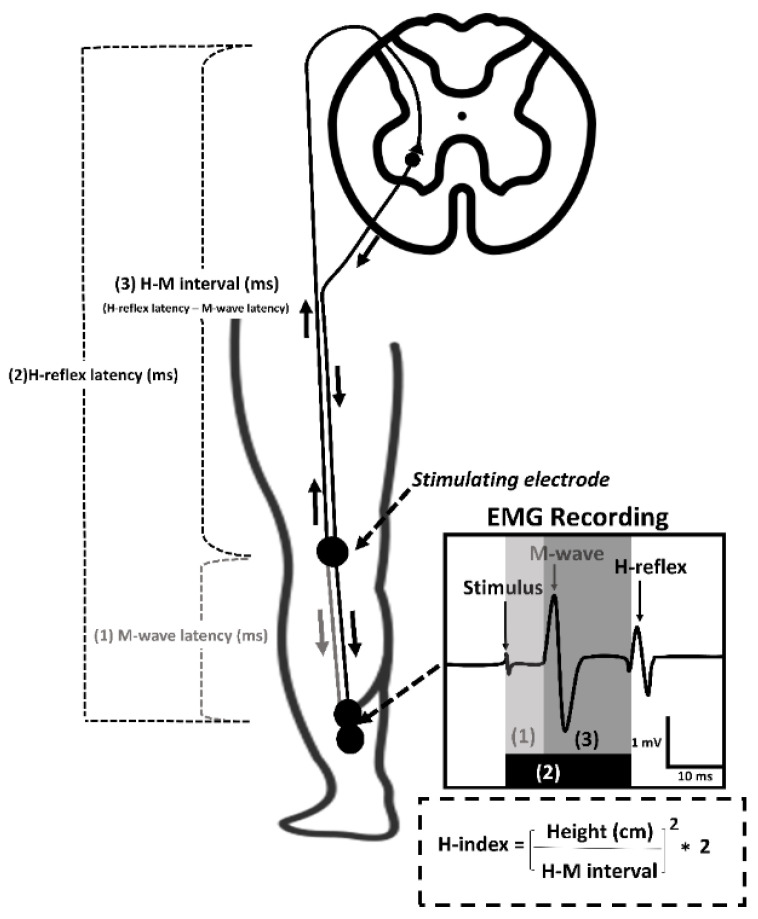
Analysis of H-index of lower leg muscles. (**1**) M-wave latency is a time delay between the stimulus artifact and the onset of the first wave (M-wave), reflecting the nerve conduction time of the direct efferent pathway. (**2**) H-reflex latency is a time delay between the stimulus artifact and the onset of the second wave (H-reflex), reflecting the nerve conduction time of both afferent and efferent pathways through the spinal reflex arc. (**3**) H-M interval is a time interval between the onset of M-wave and H-reflex, reflecting the conduction time of both afferent and efferent pathways, excluding the direct efferent pathway. A formula for H-index calculation provides normalization relative to participant height (dotted box, bottom right).

**Figure 2 healthcare-10-01794-f002:**
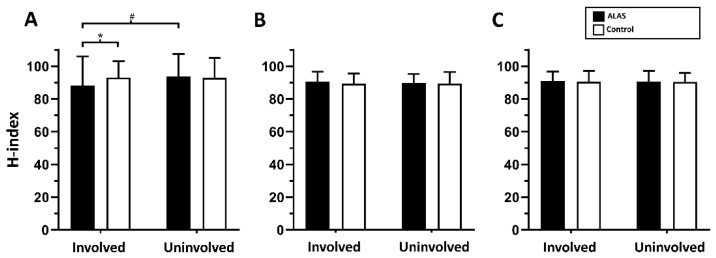
Comparison of H-index of the involved and uninvolved limb between two groups. Group data for soleus (**A**), fibularis longus (**B**), and tibialis anterior (**C**). * Soleus H-index of the involved limb in the acute ankle sprain group is significantly slower than that in the control group, *p* < 0.05. ^#^ Soleus H-index of the involved limb is significantly slower than that of the uninvolved limb in the acute ankle sprain group, *p* < 0.05.

**Figure 3 healthcare-10-01794-f003:**
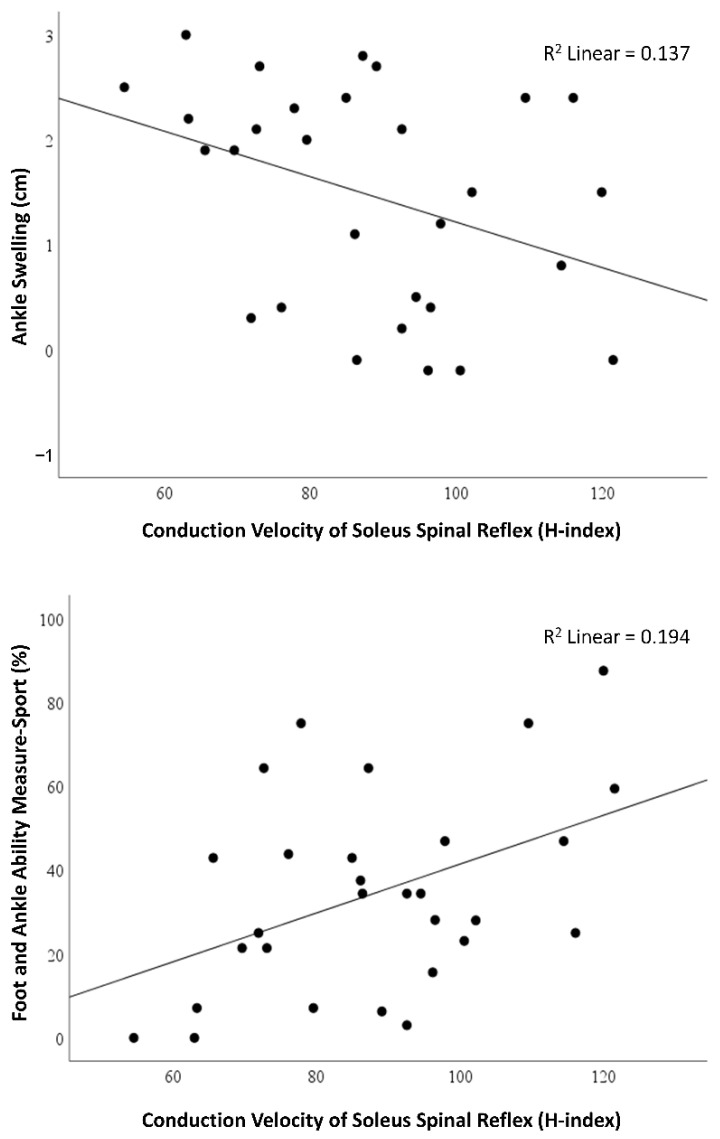
Relationship between the soleus H-index and acute symptoms: ankle swelling (**top**) and self-reported ankle function during sports activities (**bottom**) in patients with ALAS.

**Table 1 healthcare-10-01794-t001:** Descriptive data of H-reflex and M-wave latencies.

Muscle	Latency (ms)	ALAS		Control	
		Involved	Uninvolved	Involved	Uninvolved
Soleus	H-reflex latency	36.5 ± 3.9 *^†^	35.4 ± 3.3	35.3 ± 3.6	35.3 ± 3.6
	M-wave latency	9.7 ± 1.1	9.6 ± 1.1	9.7 ± 1.6	9.7 ± 1.7
Fibularis longus	H-reflex latency	31.7 ± 2.4	32.0 ± 2.4	31.5 ± 2.0	31.6 ± 1.9
	M-wave latency	5.7 ± 0.9	5.7 ± 0.9	5.6 ± 0.7	5.7 ± 0.7
Tibialis anterior	H-reflex latency	32.9 ± 2.0	33.0 ± 1.9	33.0 ± 2.6	33.0 ± 2.6
	M-wave latency	6.8 ± 0.8	6.8 ± 0.9	7.0 ± 1.3	7.0 ± 1.3

ALAS, acute lateral ankle sprain. A significant difference was found in H-reflex latency of the involved limb compared with the contralateral, uninvolved limb of the ALAS group (*) and the side-matched limb in the healthy control group (^†^).

**Table 2 healthcare-10-01794-t002:** Pearson correlation coefficients between the conduction velocity of the soleus spinal reflex and acute symptoms in ALAS group.

	Soleus H-Index ^a^	
	*r* (95% Confidence Interval)	*p*
Ankle swelling (cm)	−0.371 * (−0.649 to −0.005)	0.048
VAS score for pain (cm)	0.148 (−0.231 to 0.488)	0.443
FAAM-ADL (%)	0.167 (−0.213 to 0.503)	0.387
FAAM-Sport (%)	0.440 * (0.088 to 0.695)	0.017

^a^ Data from the involved limb of the acute lateral ankle sprain group. * Indicates a significant correlation.

## Data Availability

Not applicable.

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
