# Peer review of "Conduction Velocity of Spinal Reflex in Patients with Acute Lateral Ankle Sprain"

_healthcare, 2022, doi:10.3390/healthcare10091794_

Round 1
Reviewer 1 Report
General comments: The manuscript presented addresses a relevant research topic. It addresses the conduction velocity of spinal reflex pathways following an acute lateral ankle sprain. It is a secondary analysis of data from a previously published study though an interesting topic and well designed.
Introduction
Is presented satisfactorily
Methods
It is satisfactorily written although concerning the participants, if possible, it should clarify that both legs were assessed in the control group and if the lower limb dominance was controlled. Please define the dominance used.
Results
Participant characteristics, that are presented in this topic, should include the injured participants' characteristics without having to consult another paper. Age, weight and height are mentioned in the topic of the methods.
Table 1. : although in the caption it is written “side matched limb” it does not seem right to write injured in the control group. Please correct.
On the other hand, I question the rationale for using the side-matched limb and not the dominant and non-dominant matching. This issue should be addressed in the introduction.
Discussion
It is presented satisfactorily.
Conclusion
It is presented satisfactorily. However, it would be improved if it includes the clinical implications of the study's findings.
Author Response
We would like to thank the editor and reviewers for your thorough review and observations to improve this manuscript. We have addressed all the comments, which would help strengthen the quality of the manuscript.
|
REVIEWER 1 |
|
Material and Methods |
|
Comment 1: It is satisfactorily written although concerning the participants, if possible, it should clarify that both legs were assessed in the control group and if the lower limb dominance was controlled. Please define the dominance used. Author Response: We added clarification “The H-reflex and M-waves were measured bilaterally …” at line 147. The influence of limb dominance was controlled by performing the side-matching procedure. |
|
Results |
|
Comment 2: Participant characteristics, that are presented in this topic, should include the injured participants' characteristics without having to consult another paper. Age, weight and height are mentioned in the topic of the methods. Author Response: We added participant characteristics in lines 189-193 in line with the reviewer’s recommendation. |
|
Comment 3: Table 1. : although in the caption it is written “side matched limb” it does not seem right to write injured in the control group. Please correct. Author Response: All relevant words were corrected to “involved” and “uninvolved” throughout the entire manuscript. |
|
Comment 4: On the other hand, I question the rationale for using the side-matched limb and not the dominant and non-dominant matching. This issue should be addressed in the introduction.
Author Response: Side matching was performed in order to prevent any influence of limb dominance on our H-reflex measures so that proportion of right and left limbs in both groups can be equally balanced. The rationale with a reference was added at line 148. |
|
Conclusion |
|
Comment 5: It is presented satisfactorily. However, it would be improved if it includes the clinical implications of the study's findings. Author Response: We included the clinical implications in line 373
|

Reviewer 2 Report
General Comments
The topic of research is of interest as ankle sprains if not treated properly, lead to chronic ankle instability in which neural conduction and proprioception have an important role
It is written on an adequate standard of English language.
Abstract
It is concise and well structured. Content is adequate as it mentions what was done, the results and conclusion.
Level of evidence should be specified.
Title
It is concise and short
Introduction
It mentions the necessary background. It explains the purpose of the study and the hypothesis is specified
Methods and materials
The method description is accurate. It allows to make this study replicable with the level of detail in the methods section
Was there a sample size calculation performed? This is particularly important if groups were compared and no difference was found. Was this because there is no difference or because the sample size is low?
Line 107: Could you clarify how was the ligaments injury established? Was MRI used in each patient?
Results
It is consistent with the methods. Numbers are reported adequately. Good use of tables and figures.
Line 212 no difference in efferent conduction (M-wave) when comparing the different cohorts. Please develop a paragraph of discussion regarding that finding. There is no mention of any efferent conduction within the discussion. We believe that is important because the afferent conduction as you showed, is affects in ALAS, but if the efferent reaction has no difference, does that mean that injured and uninjured would have the same efferent response?
Discussion
Correct mention of the main findings of the study and mention of clinical relevance.
Needs discussion related to M-wave findings.
Conclusion
It is short, concise and based on the findings of the study
Author Response
We would like to thank the editor and reviewers for your thorough review and observations to improve this manuscript. We have addressed all the comments, which would help strengthen the quality of the manuscript.
|
REVIEWER 2 |
|
Abstract |
|
Comment 1: It is concise and well structured. Content is adequate as it mentions what was done, the results and conclusion. Level of evidence should be specified. Author Response We specified Level of Evidence in line 31 in line with the reviewer’s recommendations. |
|
Material and Methods |
|
Comment 2: The method description is accurate. It allows to make this study replicable with the level of detail in the methods section
Was there a sample size calculation performed? This is particularly important if groups were compared and no difference was found. Was this because there is no difference or because the sample size is low? Author Response: An a priori power analysis was conducted to estimate the sample size required for the current study. Based on a previous H-reflex study (Klykken et al. Motor-Neuron Pool Excitability of the Lower Leg Muscles After Acute Lateral Ankle Sprain. J. Athl. Train. 2011, 46, 263–269), 15 participants would be needed per group to determine a significant difference with the level of 0.05 and the statistical power of 0.80. We doubled the minimal sample size to 30 participants per group to provide conclusive evidence, which is a low sample size in terms of the population (acute ankle sprain) and a priori power analysis result. According to your suggestion, we added a brief statement in the statistical analysis section (line 174). |
|
Comment 3: Line 107: Could you clarify how was the ligaments injury established? Was MRI used in each patient? Author Response: MRI was not used in patients with ankle sprains. A certified and licensed athletic trainer evaluated all patients with acute ankle sprains by using a standardized ankle injury evaluation form. The ankle injury evaluation form consisted of an assessment of ankle swelling, palpation, special manual tests, and other functional evaluations in order to diagnose the injury. Acute ankle sprain was considered if at least minimal pain/ tenderness, swelling, and loss of function were present in individuals with acute ankle sprains. This has been noted in the manuscript on Line 117. |
|
Results |
|
Comment 4: It is consistent with the methods. Numbers are reported adequately. Good use of tables and figures.
Line 212 no difference in efferent conduction (M-wave) when comparing the different cohorts. Please develop a paragraph of discussion regarding that finding. There is no mention of any efferent conduction within the discussion. We believe that is important because the afferent conduction as you showed, is affects in ALAS, but if the efferent reaction has no difference, does that mean that injured and uninjured would have the same efferent response? Author Response: Based on your valuable suggestion, we developed and added a paragraph regarding M-wave latency in line 297. |
|
Discussion |
|
Comment 4: Correct mention of the main findings of the study and mention of clinical relevance. Needs discussion related to M-wave findings Author Response: As mentioned above, we discussed M-wave findings in the paragraph beginning on line 297
|
